# Tolerance Induced by Antigen-Loaded PLG Nanoparticles Affects the Phenotype and Trafficking of Transgenic CD4^+^ and CD8^+^ T Cells

**DOI:** 10.3390/cells10123445

**Published:** 2021-12-07

**Authors:** Tobias Neef, Igal Ifergan, Sara Beddow, Pablo Penaloza-MacMaster, Kathryn Haskins, Lonnie D. Shea, Joseph R. Podojil, Stephen D. Miller

**Affiliations:** 1Department of Microbiology-Immunology, School of Medicine, Northwestern University Feinberg, 303 E. Chicago Avenue, Chicago, IL 60611, USA; tobias.neef@northwestern.edu (T.N.); igal.ifergan@northwestern.edu (I.I.); sara.beddow@northwestern.edu (S.B.); ppm@northwestern.edu (P.P.-M.); j-podojil@northwestern.edu (J.R.P.); 2Department of Immunology and Microbiology, School of Medicine, University of Colorado, Aurora, CO 80045, USA; katie.haskins@cuanschutz.edu; 3Department of Biomedical Engineering, University of Michigan, Ann Arbor, MI 48109, USA; ldshea@umich.edu; 4Research & Development, Cour Pharmaceuticals Development Company, Northbrook, IL 60062, USA

**Keywords:** tolerance, nanoparticles, type 1 diabetes, cell trafficking, chemokine receptors, regulatory T cells

## Abstract

We have shown that PLG nanoparticles loaded with peptide antigen can reduce disease in animal models of autoimmunity and in a phase 1/2a clinical trial in celiac patients. Clarifying the mechanisms by which antigen-loaded nanoparticles establish tolerance is key to further adapting them to clinical use. The mechanisms underlying tolerance induction include the expansion of antigen-specific CD4^+^ regulatory T cells and sequestration of autoreactive cells in the spleen. In this study, we employed nanoparticles loaded with two model peptides, GP_33–41_ (a CD8 T cell epitope derived from lymphocytic choriomeningitis virus) and OVA_323–339_ (a CD4 T cell epitope derived from ovalbumin), to modulate the CD8^+^ and CD4^+^ T cells from two transgenic mouse strains, P14 and DO11.10, respectively. Firstly, it was found that the injection of P14 mice with particles bearing the MHC I-restricted GP_33–41_ peptide resulted in the expansion of CD8^+^ T cells with a regulatory cell phenotype. This correlated with reduced CD4^+^ T cell viability in ex vivo co-cultures. Secondly, both nanoparticle types were able to sequester transgenic T cells in secondary lymphoid tissue. Flow cytometric analyses showed a reduction in the surface expression of chemokine receptors. Such an effect was more prominently observed in the CD4^+^ cells rather than the CD8^+^ cells.

## 1. Introduction

Polymer-based nanoparticles are being explored as platforms for immunotherapy. Examples include treatment for type 1 diabetes (T1D) [1], other forms of antigen-specific therapy [2], and as cancer vaccines [3,4]. We have produced carboxylated poly(lactide-*co*-glycolide) (PLG) nanoparticles that induce immune tolerance in a wide variety of mouse disease models in a manner partly dependent on CD4^+^ regulatory T cells (Tregs) [5,6,7,8]. Initially, polystyrene beads coated with myelin proteolipid protein (PLP) 139–151 were shown to induce tolerance in mouse experimental autoimmune encephalomyelitis (EAE). The tolerance was mainly reversed by the administration of anti-IL-10 or anti-CD25 antibodies, suggesting that CD4^+^CD25^+^FoxP3^+^ Tregs were in part responsible for establishing tolerance [5]. Later, antigen (Ag)-loaded biodegradable carboxylated (Ag-PLG) nanoparticles, a more clinically translatable tolerance delivery system, were shown to prevent and treat disease in several mouse models of T1D. The treatment with particles encapsulating or coated with the chromogranin A mimotope peptide, P31, following the transfer of diabetogenic transgenic CD4^+^ BDC-2.5 T cells to NOD.*scid* mice led to their sequestration in the spleen and increased CD4^+^FoxP3^+^ Tregs in the spleen and pancreas [6]. These cells were found to be important for long-term tolerance as the removal of CD25^+^ T cells drastically reduced the time through which tolerance was maintained. Our more recent studies in the BDC-2.5 CD4^+^ T cell adoptive transfer model showed that PLG nanoparticles coupled with a hybrid insulin peptide, 2.5HIP, increased the numbers of FoxP3^+^ Tregs in the pancreas and reduced the trafficking of effector T cells [7]. The diminished expression of the Th1 regulator T-bet, which affects the expression of the chemokine receptor (CKR) CXCR3, may have contributed to this altered trafficking. Finally, gliadin-loaded PLG nanoparticles led to increased FoxP3 mRNA expression in a mouse model of celiac disease [8].

Tolerance studies have largely focused on CD4^+^ Tregs [9,10,11,12], but CD8^+^ Tregs have also been described [13,14]. We have reported that tolerance can be induced in NOD.*scid* mice that receive transgenic diabetogenic CD8^+^ NY8.3 T cells by using particles encapsulating the IGRP mimotope peptide NRP-A7, suggesting that Ag-PLG nanoparticles are also able to expand CD8^+^ Tregs [6]. The cell surface markers associated with CD8^+^ Tregs have been broadly grouped into three subsets: CD8^+^FoxP3^+^, CD8^+^CD28^−^, and CD8^+^CD122^+^ [13,14,15,16,17,18]. Since CD122 has traditionally been associated with CD8^+^ memory T cells [13], a third marker, e.g., PD-1, has often been proposed to further define CD8^+^ Tregs. The co-transfer of CD8^+^CD122^+^PD-1^+^, but not CD8^+^CD122^+^PD-1^−^ T cells, along with effector T cells, inhibited BALB/c to Rag1^−/−^ skin graft rejection in a manner that was partially dependent on IL-10 [19]. In another study, CD8^+^CD122^+^PD-1^+^ T cells enriched by dendritic cell AIF1 silencing were able to suppress the proliferation of transgenic CD4^+^ OT-II cells in vitro and in vivo, again partially dependent on IL-10 [20]. CD8^+^ Tregs have also been described in another model of graft rejection in which BALB/c heart grafts in CD8^−/−^ mice were protected by a cell contact-independent mechanism by the transfer of CD8^+^ T cells from mice given ICOS-B7h blockade, a treatment that increases the expression of PD-1 on CD8^+^ T cells, although the expression of CD122 was not examined [21]. Other studies have focused on the marker Ly49. Kim et al. found that CD8^+^CD122^+^CD44^+^Ly49^+^ T cells were able to suppress follicular helper T cells [T_FH_] and their ability to stimulate antibody production and linked a defect in their suppressive ability to a lupus-like disease in mice [22]. Yu et al. used the transfer of CD8^+^CD122^+^ cells to mitigate the severity of EAE in mice that had been depleted of these cells by anti-IL-15 antibody, noting that the cells in question expressed high amounts of both PD-1 and Ly49A [23].

An interesting comparison between these two markers can be found when examining the role of Tregs in non-obese diabetic (NOD) mice. It was found that CD8^+^CD122^+^ T cells in NOD mice actually promote diabetes development, most likely due to the decreasing percentage of them that are PD-1^+^, with increasing age, as compared with C57BL/6, leaving most of the CD8^+^CD122^+^ pool to be dominated by PD-1^−^ effectors [24]. On the other hand, the inadequate regulatory activity of NOD CD8^+^CD122^+^ T cells may be due to a lack of Ly49^+^ cells as they were determined to be deficient in this marker and unable to suppress T_FH_ stimulated antibody production in a NOD to NOD.RAG transfer system [25].

A third marker that has been used to differentiate CD8^+^ Tregs from CD8^+^ memory T cells is CD38. Memory-like cells of the CD8^+^CD122^+^CD38^+^ phenotype showed suppressive and regulatory capability both in vivo and in vitro and appear to be dependent on both IFN-γ and cell-to-cell contact [26].

Here, we have studied the effect of antigen-loaded nanoparticles on the phenotype and trafficking of transgenic CD4^+^ and CD8^+^ T cells. We show that nanoparticles loaded with their cognate antigen, LCMV GP_33–41_, expand the CD8^+^CD122^+^ Tregs in CD8^+^ transgenic P14 mice. The expanded population is double positive for CD122 and all three Treg markers mentioned above and the cells proliferate less in response to cognate antigen. Furthermore, the expanded cells show regulatory properties as CD8^+^ cells from treated mice suppress CD4^+^ cell viability in ex vivo cultures. We also found nanoparticle treatment induced an increase in CD8^+^ T cells with an exhausted LAG-3^+^ phenotype, similar to the results reported in a recent phase two trial in patients at risk for development of T1D following treatment with teplizumab [27]. We also demonstrate that both CD8^+^ P14 (GP_33–41_-specific) and CD4^+^ DO11.10 (OVA_323–339_-specific) transgenic T cells are sequestered in secondary lymphoid tissue following treatment with PLG nanoparticles loaded with their respective cognate antigen. In the BDC-2.5 transfer model, we show that PLG-p31 nanoparticle-induced IL-10-producing Tregs play an important role in the regulation of T1D. The flow cytometric analysis revealed that this may also be due to the downregulation of certain CKRs on the surfaces of cells. The continued clarification of the exact nature of the effects of Ag-PLG on CD4^+^ and CD8^+^ T cells will be important for understanding their mechanism of action and for clinical translation.

## 2. Materials and Methods

### 2.1. Mouse Strains

P14 mice were generously provided by the Penaloza-MacMaster Laboratory at Northwestern University. DO11.10 were bred at Northwestern University. C57BL/6J, BALB/c, NOD, and NOD.*scid* mice were purchased from The Jackson Laboratory (Bar Harbor, ME, USA).

### 2.2. Peptides

LCMV GP_33–41_ (KAVYNFATM) (referred to as GP33) was purchased from Anaspec (Fremont, CA, USA). OVA_323–339_ (ISQAVHAAHAEINEAGR) (referred to as OVA323) and P31 (YVRPLWVRME) were produced by the Northwestern University Simpson Querrey Institute Peptide Synthesis Core (Chicago, IL, USA). MOG_35–55_ (MEVGWYRSPFSRVVHLYRNGK) (referred to as MOG35) was purchased from Genemed Synthesis (Genemed Synthesis, San Antonio, TX, USA).

### 2.3. Nanoparticle Synthesis and Characterization

Three different types of nanoparticles were created. Antigen-free nanoparticles, particles containing P31, or particles containing MOG35 were created via either single or double emulsion with the solvent evaporation technique described previously [5,28,29]. All other antigen-loaded nanoparticles were created by first coupling antigenic peptides to PLG polymer. Lyophilized peptide was added to 40% PLG in dimethyl sulfoxide (DMSO). Next, we created a 100 mg/mL solution of ethylene carbodiimide (ECDI) in a solvent that was one part water, one part DMSO, and added 200 μL of it to the PLG in DMSO solution, allowing the coupling reaction to incubate at room temperature for one hour under mild shaking. 50 mL of nanopure water was added and the polymer was precipitated by centrifugation at 15,000× *g* for 20 min. Following three additional washes at 3000× *g* for 5 min, the polymer was frozen at −80 °C and freeze-dried overnight. Typically, this yielded 70 to 130 mg of peptide-coupled polymer, which was mixed with uncoupled polymer to reach 400 mg total polymer mass. The single emulsion solvent evaporation procedure was then carried out.

Nanoparticle size distribution and zeta potential were determined by dynamic light scattering. Reconstituted nanoparticles were washed and finally suspended at 12.5 mg/mL in phosphate-buffered saline (PBS), then diluted in nanopore water to 0.25 mg/mL. They were then measured on a Zetasizer Nano ZS from Malvern Instruments (Westborough, MA, USA). Antigen load was determined via CBQCA assay as detailed previously [30]. 4 μL of the same PLG nanoparticle suspension was dried on a copper grid and particle morphology examined by transmission electron microscopy (TEM) with an FEI Tecnai Spirit G2 (Hillsboro, OR, USA).

### 2.4. Nanoparticle Treatment

Reconstituted nanoparticles were washed and finally suspended at 12.5 mg/mL in sterile PBS. Mice were treated with 200 μL via intravenous injection. In the case of multiple injections, nanoparticles were given once daily for a period of 5 consecutive days.

### 2.5. In Vitro Culture for Secreted Cytokine and Proliferation

For each in vitro cell culture condition, 0.2 × 10^6^ cells (either whole spleen or whole lymph nodes cells) were resuspended in 150 μL of complete RPMI medium and plated in flat-bottom 96-well plates with 1 μg/mL of GP33, anti-CD3 antibody (145-2C11), or both and allowed to proliferate at 37 °C for the specified time period. Amounts of soluble cytokines from recall cultures was determined by centrifuging well plates at 1700 RPM for 3 min and removing the supernatant. 25 μL of each well was analyzed by Luminex magnetic bead assay according to the manufacturer’s instructions. Analytes were IL-4, IL-10, IL-17, GM-CSF, IFN-γ, and TNF-α (MilliporeSigma, Burlington, MA, USA). For determining proliferation, cells were pulsed with 1 μCi/well [^3^H]TdR on day 2. Cell cultures were harvested on day 3 and proliferation was determined via scintillation using a Topcount microplate reader (PerkinElmer, Waltham, MA, USA).

### 2.6. Magnetic Cell Separation

CD122^+^ cells were positively selected and removed from whole spleens by first staining with biotin conjugated anti-CD122 monoclonal antibody and then removing the stained cells with a biotin positive selection kit (STEMCELL Technologies, Vancouver, BC, Canada) according to the manufacturer’s instructions.

### 2.7. CFSE Dilution

For proliferation experiments using carboxyfluorescein succinimidyl ester (CFSE) dilution, 8 × 10^6^ cells/mL were stained before incubation with 5 mM CFSE-Vi (Thermo Fisher Scientific, Waltham, MA, USA) following the recommended protocol. Amount of proliferation was quantified as “proliferation score” using the method outlined by Ahlen et al. [31].

### 2.8. Flow Cytometry

Prior to staining, spleens and/or brachial, axillary, mesenteric lymph nodes were harvested and single cell suspensions were created. After red blood cell lysis and washing, cells were incubated with Fc block for 20 min, washed, and stained with live/dead stain for 20 min. Following additional washing, cells were stained with monoclonal antibodies at the recommended concentrations for 20 min. In the case of intracellular staining, cells were also washed with fixation/permeabilization solution and buffer. Antibody specificities include: CD3, CD4, CD8, CD122, CD28, FoxP3, PD-1, Ly49, CD38, TIGIT, KLRG1, LAG-3, DO11.10, CD45.1, CCR1, CCR2, CCR4, CCR5, CCR6, CCR7, CCR8, CXCR2, CXCR3, CXCR4, CXCR5, CXCR6. All antibodies were from BioLegend (San Diego, CA, USA) and eBiosciences (Waltham, MA, USA).

### 2.9. BDC-2.5 Adoptive Transfer

Spleens and brachial, axillary, mesenteric, and pancreatic lymph nodes were harvested from transgenic BDC-2.5 mice. The cells were cultured in complete RPMI at 37 °C with 0.5 μM P31 in round-bottom 96-well plates for 96 h. After washing, the cells were resuspended in PBS and intravenously injected into NOD.*scid* recipients (10 × 10^6^ per mouse). The NOD.*scid* mice were treated with either PLG(P31) or PLG(MOG35) within 24 h following cell transfer, and their blood glucose levels were monitored. Mice with blood glucose over 250 mg/dL were considered diabetic and removed from the study. Starting at day 31 after transfer, mice that had been treated with PLG(P31) received daily intraperitoneal antibody, either monoclonal anti-IL-10 (JES5-2A5) (BioXCell, Lebanon, NH, USA) or isotype control, until day 40 after transfer.

### 2.10. Serum Cytokines/Chemokines

Mice were bled and, after 30 min of room temp incubation, samples were centrifuged and serum was collected. 12.5 μL from each mouse was analyzed by Luminex magnetic bead assay according to the manufacturer’s instructions. Standard cytokine analytes were IL-6, IL-4, IL-10, IL-17, IFN-γ, and TNF-α. Chemokine analytes were MCP-1/CCL2, MIP1α/CCL3, MIP-1β/CCL4, KC/CXCL1, and MIP-2/CXCL2.

### 2.11. Statistics

All statistical analyses were performed using GraphPad Prism 7.0 software. *p* < 0.05 was considered significant.

## 3. Results

### 3.1. Nanoparticle Synthesis and Characterization

The nanoparticles used in this study, as well as their individual characteristics, are displayed in Table 1. Representative TEM images are displayed in Figure 1. The PLG particles without any antigen were produced by single emulsion. The properties of three separate preparations are displayed to demonstrate the typical variation between the different batches. Particles loaded with the MHC class I-restricted GP_33–41_ (PLG/GP33) or the MHC class II-restricted OVA_323–339_ (PLG/OVA323) peptides were produced by the coupling of the peptide to the PLG polymer followed by single emulsion. Particles loaded with BDC-2.5 mimetope 1040-31 (PLG(P31)) and the irrelevant control peptide MOG_35–55_ (PLG(MOG35)) were produced by double emulsion. The diameter and zeta potential are similar across all five types, and the differences between them are normal batch-to-batch variations. Differences, however, were found in terms of the amount of antigen loaded into each type of particle. PLG/GP33 only carries 3.4 μg peptide per mg of nanoparticles, whereas PLG/OVA323 has a load of 64.5 μg/mg of nanoparticles. The reason for the differences in the antigen load between the different antigens, even when using the same nanoparticle synthesis technique, remains to be discovered. Confirming that each batch of nanoparticles contains an adequate amount of antigen by CBQCA assay is critical to ensuring they exert an immunological effect. Release profiles of the antigens, as well as other properties, were found for similar particles in previous studies [32,33].

### 3.2. Antigen-Encapsulating PLG/GP33 Nanoparticles Expand CD8^+^ Tregs in P14 Transgenic Mice

Our previous results [6,7,8] have shown that antigen-bearing biodegradable PLG nanoparticles induce the expansion of CD4^+^FoxP3^+^ Tregs in various disease models; therefore, we injected PLG/GP33 or control PLG (PLG/OVA323 or PLG) intravenously into female transgenic P14 mice to determine if the treatment altered the antigen-specific CD8^+^ T cell phenotype, similar to antigen-specific CD4^+^ T cells (Figure 2). A flow cytometric analysis of the CD8^+^ T cells was carried out to quantify the expression of potential Treg markers (Figure 2A). The expression of FoxP3 was not affected to a significant degree following the PLG/GP33 treatment.

In contrast, both CD28 and CD122 showed an increase in mean fluorescence intensity (MFI). To further distinguish if the PLG/GP33 treatment induced a regulatory phenotype in the resultant CD8^+^CD122^+^ T cells, we examined the expression of PD-1, Ly49, and CD38 (Figure 2B). The present data show that the populations of all three of these markers significantly increased following the PLG/GP33 treatment. Due to a report that the expression of CD122 among CD8^+^ T cells varies with age [34], both the PLG/GP33 and control particles were injected into three mice of different ages: young (4 weeks), adult (8 weeks), and old (12 weeks) (Figure 2C). The present data show that the age-related differences were small compared to the treatment the mice received.

### 3.3. Recall Culture of PLG/GP33-Treated Splenocytes and Lymphocytes

We next sought to understand how the expansion of CD8^+^CD122^+^ Tregs may affect the ex vivo recall responses of splenic or lymph node CD8 T cells from PLG/GP33-treated P14 mice upon stimulation with cognate antigen. The cells from treated mice proliferated significantly less than those from mice treated with control particles (Figure 3A). This pattern was found in both spleen and lymph node cultures but was only statistically significant in the former.

Additionally, we quantified cytokines in the supernatants of the cultures. Generally, the cells from the PLG/GP33-treated mice secreted fewer cytokines than those from control-treated mice (Figure 3B). For spleen cell cultures, this difference was statistically significant for IL-10, IL-17, IL-4, GM-CSF, and TNF-α. For lymph nodes, the decrease was significant only for GM-CSF.

### 3.4. Expanded CD8^+^CD122^+^ Tregs Reduce Viability of CD4^+^ Cells

To determine if the expansion of the CD8^+^CD122^+^ Tregs induced by nanoparticle treatment correlated with the suppression of other immune cells, we compared ex vivo splenocyte cultures from mice treated with PLG/GP33 versus control mice. Initially, a variety of conditions were tested. Splenocytes from a treated P14 transgenic or untreated wild-type mouse were stimulated with either GP33, anti-CD3 antibody, or both for 3 or 24 h. The viability of the CD4^+^ T cells within the culture was determined by flow cytometry. At 3 h, significant differences were found for all the conditions. At 24 h, the differences between the two groups were only significant when anti-CD3 was included and had smaller *p*-values generally (Figure 4A). For further experiments, ex vivo suppression by CD8^+^ Tregs was determined at the 3 h timepoint upon stimulation with anti-CD3. The splenocytes from a mouse treated with PLG/OVA323 had significantly fewer viable CD4^+^ T cells compared to a mouse treated with control PLG nanoparticles, thereby confirming that the expanded CD8^+^CD122^+^ Tregs suppress CD4^+^ T cells (Figure 4B).

To determine if the expanded CD8^+^CD122^+^ Tregs were responsible for the decreased CD4^+^ T cell viability, the CD122^+^ cells were depleted using magnetic cell separation. Consequently, the depletion of the CD122^+^ cells greatly led to a significant increase in CD4^+^ T cell viability (Figure 4C), thereby showing that the CD122^+^ cells were responsible for the decrease in CD4^+^ T cell viability.

### 3.5. Antigen-Encapsulating PLG/GP33 Nanoparticles Induce Partial Exhaustion

It was recently reported that anti-CD3 can expand CD8^+^ Tregs [25,35]. In addition, a recent trial of teplizumab in relatives at risk for developing T1D demonstrated the induction of an exhausted phenotype in CD8^+^ cells [27]. We, therefore, looked for an increase in the percentage of CD3^+^ T cells that were TIGIT^+^KLRG1^+^CD8^+^, as well as an increase in the exhaustion marker LAG-3 on CD8^+^ cells following PLG/Ag treatment. Following treatment with PLG/GP33 or control PLG (PLG/OVA323 or PLG), no significant increase in TIGIT^+^ KLRG1^+^ CD8^+^ was found (Figure 5A). However, the LAG-3 on CD8^+^ T cells was significantly increased in mice receiving PLG/GP33 nanoparticles versus those receiving control particles (Figure 5A). To determine if these cells were exhausted, they were stained with CFSE and cultured in vitro for 5 days with GP33 peptide. Following incubation, the CD8^+^ T cells had lost any significant difference in the LAG-3 expression between the two treatment groups (Figure 5B). However, when gating on LAG-3 positive or negative cells and examining CFSE dilution, we determined the LAG-3 positive cells proliferated significantly less than the LAG-3 negative cells, thereby confirming their exhausted nature (Figure 5C).

### 3.6. Tolerance in BDC-2.5 Transfer Model Is Partly Due to IL-10

Due to our recent finding that treatment with antigen-loaded nanoparticles affected the trafficking of diabetogenic BDC-2.5 T cells, leading to their sequestration in spleens [6], we sought to more clearly understand the mechanisms behind this. In an adoptive transfer model of T1D, BDC-2.5 cells were stimulated in vitro with the mimetope P31 and injected into NOD.*scid* recipients. The mice were then treated with either PLG(P31) or PLG(MOG35). The PLG(P31)-treated mice were protected from development of T1D compared to controls treated with PLG(MOG35). The tolerance in the PLG(P31)-treated mice was reversed by treatment with anti-IL-10 beginning at day 30 after transfer (Figure 6). This experiment demonstrates that tolerance induction due to antigen-loaded nanoparticles is partially mediated by regulatory cells that secrete IL-10 and may affect the trafficking of effector T cells, sequestering them in spleens.

### 3.7. Single Injection of Antigen-Loaded Nanoparticles Does Not Significantly Alter Chemokine Receptor Expression

Based on the possibility that modulated chemokine levels are responsible for the altered trafficking seen in our earlier study [6], a magnetic bead assay was used to measure the levels of specific chemokines and other cytokines in the sera of mice that received a single injection of antigen-loaded versus control nanoparticle. For P14 transgenic mice, there was no significant difference between the PLG/GP33 and control PLG treated mice (data not shown). For DO11.10, some significant differences were found between the treatment groups. However, in both cases, the increased or decreased group was control PLG; there was no difference between PLG/OVA323 and PBS (data not shown). This indicates the differences seen in serum cytokines are not due to antigen-specific tolerization.

Similar results were seen when the mean fluorescence intensities (MFI) of the cytokine receptors (CKR) were examined in the spleens of the same mice immediately following serum collection (data not shown). Significant differences were seen only in the levels of CCR6 and CCR7, but, again, this is unlikely to be due to antigen-specific tolerization as there was no consistent difference between PLG/OVA323 and both controls. Taken together, these results suggest either that the spleen sequestration seen in our earlier study was not due to chemokine-related mechanisms or that the techniques we used were not sensitive enough to measure what may be subtle differences in the chemokine or CKR levels.

### 3.8. Multiple Injections of Antigen-Loaded Nanoparticles Sequester Transgenic CD4^+^ T Cells in Secondary Lymphoid Tissue and Alter Expression of Chemokine Receptors

Although a single injection of the antigen-loaded nanoparticle altered the trafficking of transgenic T cells enough to sequester them in spleens [6], we hypothesized that a greater dose was needed to produce effects that were measurable by magnetic bead assay or flow cytometry. To increase the ratio of nanoparticle to transgenic cells, CD4^+^ T cells were isolated from DO11.10 mice and transferred to wild-type BALB/c mice, which were then given five daily nanoparticle injections. The serum chemokine levels still showed no significant differences between the treated and control groups at the end of the treatment period (data not shown).

The percentages of transgenic T cells (percentage of CD3^+^CD8^−^ DO11.10 TCR) present within the spleens of the recipient mice were determined by flow cytometry, and the percentage of DO11.10 T cells was found to be elevated in the mice treated with PLG/OVA323 relative to mice receiving either control PLG and PBS treatment (Figure 7A). A similar, although not as significant, effect was seen in the lymph nodes. These results appear to show an antigen-specific homing to secondary lymphoid tissues, primarily the spleen. The examination of the splenic DO11.10 cells showed an across-the-board decrease in the MFI of CKRs (Figure 7B). For every receptor measured, a significant difference was found between the PLG/OVA323 versus either the control PLG or PBS vehicle, while none was found between the control PLG and the vehicle. This indicates an antigen-specific effect of antigen-loaded nanoparticles causing decreased CKR expression on the surface of antigen-specific DO11.10 CD4^+^ T cells.

### 3.9. Multiple Injections of Antigen-Loaded Nanoparticles Sequester Transgenic CD8^+^ T Cells in Secondary Lymphoid Tissue and Alter Expression of Chemokine Receptors

To extend the above results to CD8^+^ T cells, the spleens from P14 mice were harvested and the CD8^+^ T cells isolated. Following the transfer to wild-type C57BL/6J mice, multiple daily injections of nanoparticle were given. The percentages of transgenic T cells (percentage of CD8^+^ positive for congenic marker CD45.1) in the spleens and lymph nodes were determined by flow cytometry. Compared to secondary lymphoid tissues from mice treated with control PLG, those from the mice treated with PLG/GP33 had significantly more transgenic T cells (Figure 8A). Examining the MFI of the CKRs on the P14 cells in the spleen showed that most CKR levels remained unchanged. However, the MFI of CCR6 was significantly increased and the MFI of CXCR6 significantly decreased by the PLG/GP33 versus control PLG treatment (Figure 8B). This apparent antigen-specific change in the CKR expression was not as profound as in DO11.10 cells, which may be due to either a difference in CD4^+^ and CD8^+^ T cells, or a difference in the peptide loading between the nanoparticles used (Table 1).

## 4. Discussion

Specific immune tolerance induced by antigen-loaded biodegradable PLG nanoparticles has proven effective for ameliorating disease progression in a number of models of autoimmunity and other disorders [5,6,7,8,28,29,32,36], and, more recently, was demonstrated to be safe and effective in a phase 1/2a clinical trial in celiac disease patients [37]. Our previous studies indicated that the long-term tolerance was largely, but not exclusively, due to the systemic induction of FoxP3^+^ CD4^+^ Tregs [5,6,7,8]. We also showed that an additional component of PLG/Ag-induced tolerance involved the sequestration of transgenic effector BDC-2.5 CD4^+^ T cells in the spleen, preventing them from trafficking to the pancreas and, thus, preventing overt T1D. We also showed that Tregs played a critical role in the long-term tolerance maintenance by mediating the splenic sequestration of effector diabetogenic cells in secondary adoptive transfer experiments [6]. Currently, we show that the effector role of Tregs in the BDC-2.5-mediated T1D is IL-10 dependent (Figure 6). The major aim of the present study was to compare the effects of antigen-bearing nanoparticles on both CD4^+^ and CD8^+^ transgenic T cells. Most notably, we found that the intravenous administration of PLG nanoparticles bearing an MHC class I-restricted peptide induced the antigen-specific expansion of CD8^+^CD122^+^ transgenic T cells possessing other regulatory markers that inhibit CD4^+^ cell viability through an as yet undefined mechanism. We re-confirmed that antigen-loaded nanoparticles affect the trafficking of transgenic CD4^+^ T cells but also discovered that transgenic CD8^+^ T cells are sequestered in secondary lymphoid organs. The sequestration of these CD8^+^ T cells in spleens may be due to multiple mechanisms, including direct effects of PLG/Ag nanoparticles on effector T cells and/or by activating induced CD8^+^ Tregs to regulate effector cell trafficking, perhaps via downregulating CKRs, as has been shown in other systems [6,38,39].

The population of CD8^+^ Tregs expanded following the PLG/Ag nanoparticle treatment had the phenotype CD8^+^CD122^+^ (Figure 2A). This leaves the possibility that many of the CD122^+^ cells were simply those that had differentiated into memory cells after encountering their cognate antigen. To determine if these CD8^+^CD122^+^ T cells were, in fact, regulatory, we also confirmed the positivity of the three markers PD-1, Ly49, and CD38 (Figure 2B). A recall culture with GP33 did appear to show reduced proliferation and reduced cytokine production (Figure 5), but whether this was due to the expanded CD8^+^ Treg population, another mechanism caused by the nanoparticle treatment, or some combination of both, is still unknown. A better way to test the regulatory capability of the expanded population was to incubate the splenocytes from treated versus control mice with anti-CD3 over a short time span, which unambiguously shows less survivability for the CD4^+^ cells co-cultured with cells from nanoparticle-treated mice (Figure 4B). Depleting the CD122^+^ cells from the cultures restored the CD4^+^ cell viability, lending strong support to the notion that the expanded CD8^+^CD122^+^ cells were responsible for CD4^+^ T cell killing (Figure 4C). Previous studies have implicated cytotoxic T lymphocyte (CTL) apoptosis induction as a potential mechanism for the CD8^+^ Treg suppression of other cells [40,41]. Whether the reduced CD4^+^ T cell viability demonstrated here is dependent on Fas/FasL or granzyme/perforin will be the subject of future studies. It is also noteworthy that adding CD122^+^ cells back into the ex vivo cultures from which they were depleted did nothing to restore the reduction in CD4^+^ T cell viability. Since we utilized biotin-conjugated anti-CD122 monoclonal antibody followed by biotin specific magnetic bead separation, we hypothesize that the complicated procedure of cell separation eliminated the CD122^+^ cells’ ability to act as regulatory cells. Future studies will make use of less disruptive separation techniques.

While it was not surprising that the antigen-loaded nanoparticles in mice did not exert the same exhaustion effect as teplizumab in humans (Figure 5A, left), the significant increase in the LAG-3^+^ CD8^+^ T cells (Figure 5A, right) was of interest. Although the role of LAG-3 in our tolerance induction system has not been studied extensively, it was previously found that the *LAG3* gene was upregulated in the BDC-2.5 CD4^+^ T effectors from mice treated with P31-loaded nanoparticles [7]. Here, we have noted that this LAG-3 induction extends to CD8^+^ T cells as well. Another recent study using glycosylated antigen found a similar increase in LAG-3^+^ CD8^+^ cells following tolerization [42]. While we could confirm that LAG-3^+^ cells responded reluctantly to GP33 antigen compared to their LAG3- counterparts, no difference was observable between the treated and control groups (Figure 5C), and, in fact, the difference in the LAG-3^+^ levels disappeared over the course of the incubation (Figure 5B). Future experiments will use different time scales and different readouts of T cell response to the antigen (e.g., cytokine production) to assess the exact role that CD4^+^ and/or CD8^+^ T cell exhaustion plays in the tolerance induced by antigen-loaded nanoparticles.

The issue of spleen sequestration has been of interest to us ever since our initial study of tolerance induction in a transgenic CD4^+^ T cell transfer model of T1D [6]. It remains unclear what the mechanism behind that sequestration was, although the breaking of the tolerance using the anti-IL-10 antibody (Figure 6) suggests it may be through the action of CD4^+^FoxP3^+^ Tregs, which were found to be increased in spleens and pancreases and necessary for the maintenance of tolerance [6,7]. Here, we have shown that antigen-loaded nanoparticle treatment results in the sequestration in the secondary lymphoid tissue of both CD4^+^ (Figure 7A) and CD8^+^ (Figure 8A) T cells. The altered expression of CKRs via direct or Treg-mediated mechanisms correlates with splenic sequestration.

For CD4^+^ T cells, since every CKR that was examined exhibited decreased MFI and was potentially downregulated, it is difficult to pinpoint what the definitive effects on trafficking would be. However, two CKRs stand out. CCR7 is necessary for naïve T cells to encounter antigens in lymph nodes [43], and the downregulation of this CKR could be one arm of a multi-arm state of tolerance induced by antigen-loaded nanoparticles, although its relation to spleen sequestration is not clear. CXCR3, on the other hand, is partly responsible for the homing of Th1 CD4^+^ T cells to inflamed tissues [44,45], so the downregulation of CXCR3 would be consistent with the retention of effector CD4^+^ T cells in the spleen. Indeed, our previous results include a decrease in the expression of T-bet, a regulator of CXCR3 expression following treatment with antigen-loaded nanoparticles [7]. This would be consistent with the T-bet and CXCR3 downregulation found by Wan et al. using a mimetope-Ig chimera in a BDC-2.5 adoptive transfer [46]. While it is difficult to conjecture how CCR6 upregulation could be involved in spleen sequestration or even broader tolerance, CXCR6 has been noted to be involved in the T cell (including CD8^+^) trafficking to infected or post-vaccination tissues [47,48,49], and its downregulation could conceivably lead to spleen retention.

Several complicating factors keep us from definitively concluding that the CKR decreases observed in the current study are responsible for spleen sequestration. First, the five-day nanoparticle regime used in the current study differs from the single nanoparticle injection used in the BDC-2.5 transfer model. While it is possible that the effect of both is the same and flow cytometric techniques are not sensitive enough to detect CKR expression differences that nonetheless exert an effect biologically, we must consider the possibility that the five times larger dose leads to a different effect. Support for the latter possibility is the fact that there is no inflamed tissue to sequester the T cells away from in the cases of P14 or DO11.10 cells, unlike the treatment of diabetogenic T cells upon the transfer to NOD.*scid* recipients. It is, therefore, possible that the mechanisms regulating T cell trafficking will be different in the two different contexts. Secondly, the lower MFI of the CKRs could be due to the blocking of fluorescent antibodies from those markers by the ligation of the chemokines. It may be that antigen-loaded nanoparticles induce an increase in chemokine production, even though the differences in the levels of these between the treated and control groups were imperceptible by magnetic bead assay. Finally, while antigen-loaded nanoparticle treatment led to a broad decrease in the CKR MFI on CD4^+^ DO11.10 T cells, it only caused differences in two receptors on CD8^+^ P14 T cells. This may be due to a difference between the way CD4^+^ and CD8^+^ T cells respond to tolerogenic treatment, but it may also be caused by differences in the nanoparticles used. As shown in Table 1, PLG/OVA323 contained almost 20 times as much antigen as PLG/GP33. It appears that the Ova323 peptide coupled to the PLG polymer more effectively than GP33; the exact reason for the higher antigen load still remains unknown.

Future investigation into the effects of antigen-loaded nanoparticles on T cell trafficking will need to address these issues, as well as include more flow cytometric analyses to define which subsets of CD4^+^ and CD8^+^ T cells experience CKR modifications. It is hoped that such analyses will also be able to bridge the two themes of this paper, i.e., Tregs and T cell trafficking.

## Figures and Tables

**Figure 1 cells-10-03445-f001:**
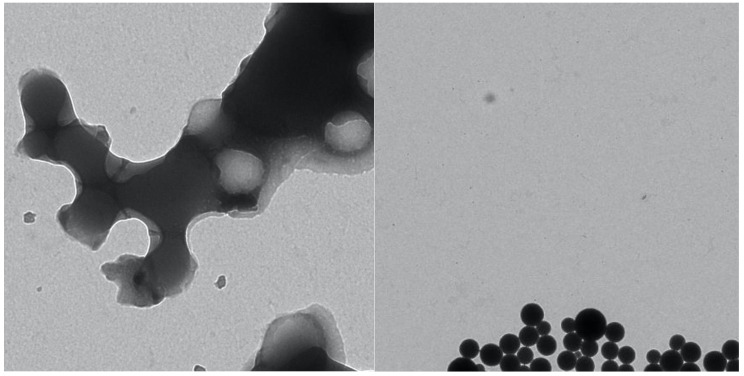
Transmission electron microscopy of PLG nanoparticles. Aqueous nanoparticle suspension was placed on a copper grid and allowed to dry, then examined with a transmission electron microscope. Images are representative of typical PLG nanoparticles.

**Figure 2 cells-10-03445-f002:**
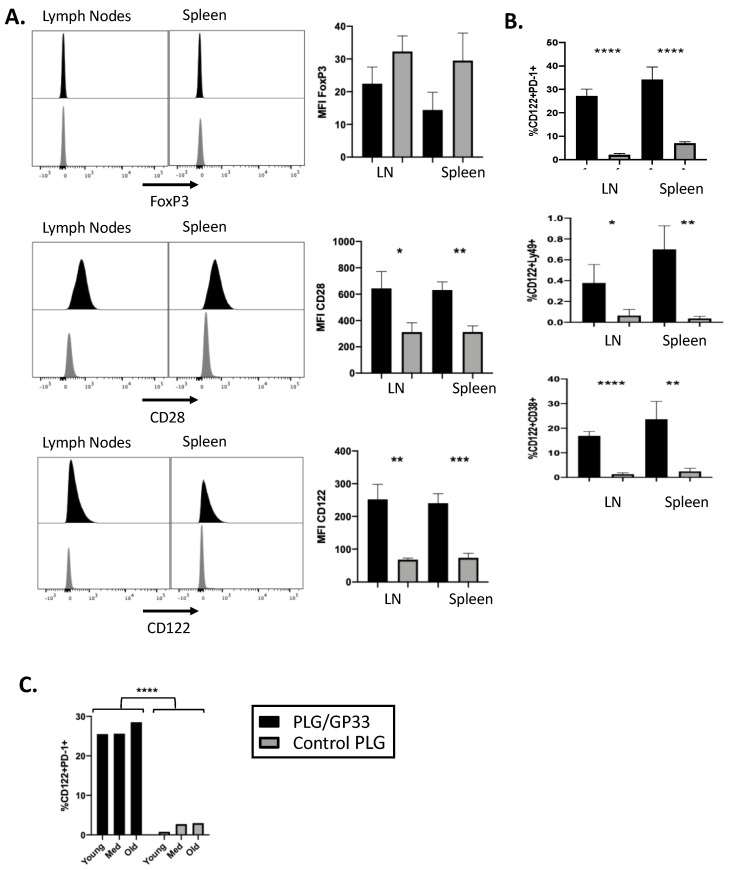
Antigen-encapsulating PLG/GP33 nanoparticles expand CD8^+^ Tregs in transgenic P14 mice. 3–4 female P14 mice were injected intravenously with either PLG/GP33 or a control PLG (PLG/OVA323 or PLG). Spleens and/or lymph nodes (pooled axillary, brachial, and inguinal) were harvested and counted 3 days later. Viable T cells were analyzed by flow cytometry. Statistical significance was determined by Student’s *t*-test. * *p* < 0.05, ** *p* < 0.01, *** *p* < 0.001, **** *p* < 0.0001. (**A**) Expression of FoxP3, CD28, or CD122 was determined by geometric mean fluorescence intensity (MFI). (**B**) Sub-populations were determined by percentage of viable CD8^+^ cells that were double positive for CD122 and PD-1, Ly49, or CD38. (**C**) Female P14 mice (*n* = 3) at various ages (young 4 weeks of age, adult 8 WOA, old 12 WOA) were injected intravenously with either PLG/GP33 or PLG/OVA323. Expansion of Tregs was determined by percentage of all viable CD8^+^ T cells that were double positive for CD122 and PD-1 by flow cytometry. Statistical significance between the groups was determined by considering three ages as three replicates.

**Figure 3 cells-10-03445-f003:**
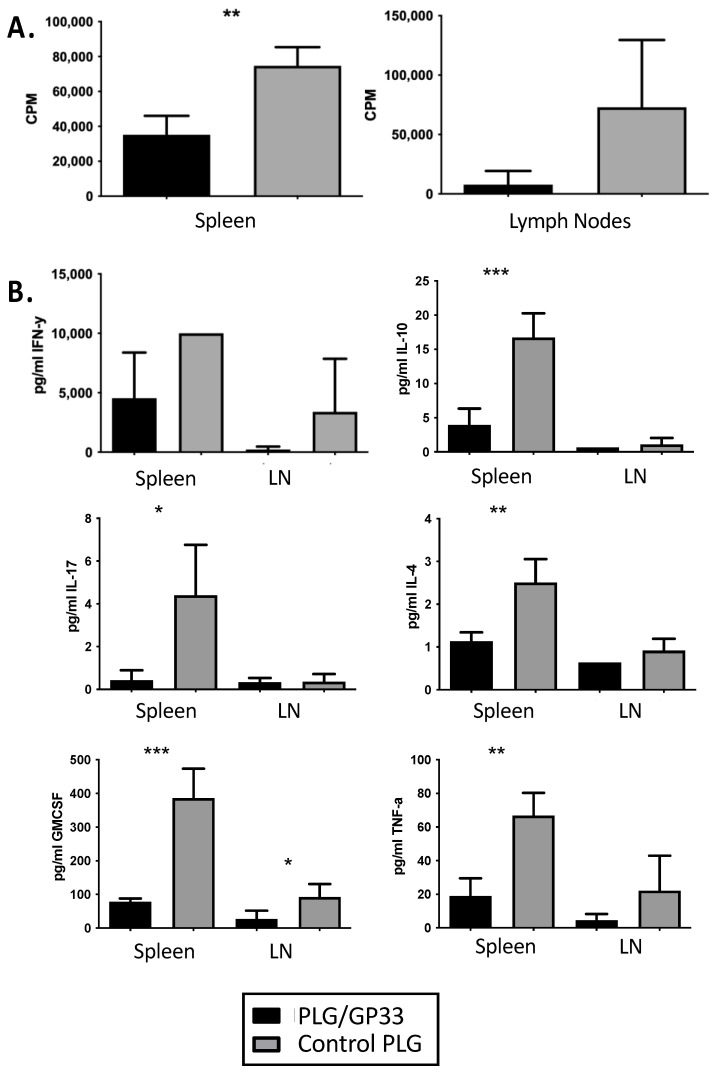
Recall culture of PLG/GP33-treated splenocytes. Female P14 mice (*n* = 3–4) were injected intravenously with either PLG/GP33 or a control PLG (PLG/OVA323 or PLG). Spleens, as well as axillary, brachial, and inguinal lymph nodes, were harvested 3 days later. Cells (2 × 10^5^ per well) were cultured in vitro with GP33 peptide for 3 days. (**A**) Proliferation was determined by [^3^H]TdR incorporation. (**B**) Cytokine levels were determined by a cytometric bead array. Statistical significance was determined by Student’s *t*-test. * *p* < 0.05, ** *p* < 0.01, *** *p* < 0.001.

**Figure 4 cells-10-03445-f004:**
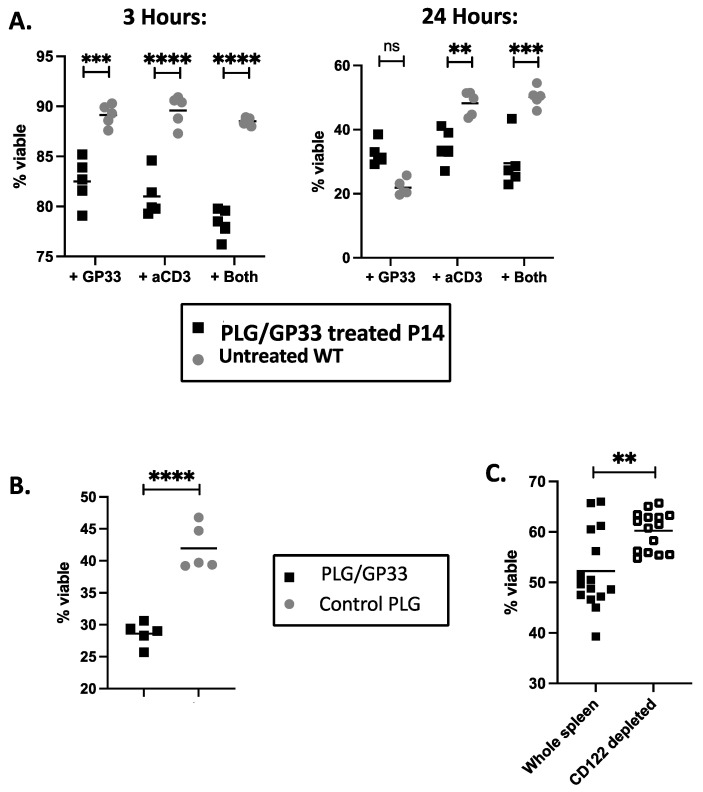
Expanded CD8^+^CD122^+^ Tregs reduce viability of CD4^+^ T cells. For each experiment, spleen cells from female P14 mice were harvested 3 days after intravenous injection with PLG/GP33 or control PLG. For each mouse, 5 separate ex vivo cultures were set up. CD4^+^ T cell viability was determined by flow cytometry. Statistical significance was determined by Student’s *t*-test. ns not significant, ** *p* < 0.01, *** *p* < 0.001, **** *p* < 0.0001. (**A**) Whole spleen cultures from PLG/GP33-treated P14 or untreated wild-type mice were incubated at 37 °C for 3 or 24 h with GP33 (1 μg/mL), anti-CD3 (1 μg/mL), or both. (**B**) Whole spleen cultures from P14 mice treated with either PLG/GP33 or control PLG (PLG/OVA323 or PLG) were incubated at 37 °C for 3 h with 1 μg/mL anti-CD3. (**C**) CD122^+^ cells from mice treated with PLG/GP33 were depleted by magnetic cell separation and incubated at 37 °C for 3 h with 1 μg/mL anti-CD3. Results are from three separate pooled experiments.

**Figure 5 cells-10-03445-f005:**
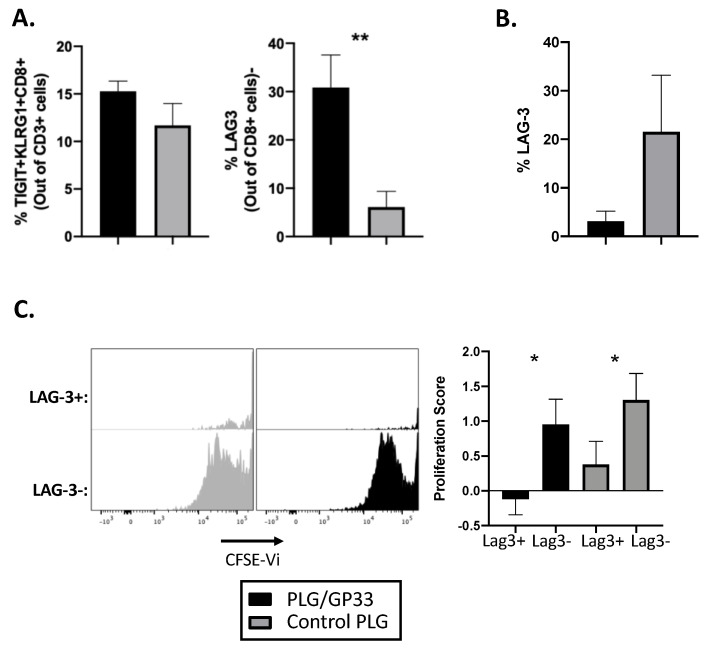
Antigen-encapsulating PLG/GP33 nanoparticles induce partial exhaustion in GP33-specific CD8^+^ T cell. Female P14 mice (*n* = 3–4) were injected intravenously with either PLG/GP33 or a control PLG (PLG/OVA323 or PLG). Spleens and/or lymph nodes (pooled axillary, brachial, and inguinal) were harvested and counted 3 days later. Viable T cells were analyzed by flow cytometry. Statistical significance was determined by Student’s *t*-test. * *p* < 0.05, ** *p* < 0.01. (**A**) Exhausted phenotype was quantified by percentage of viable CD3^+^ T cells that were triple positive for TIGIT, KLRG1, and CD8 as in Ref. [27]. LAG-3 expression was determined by percentage of viable CD8^+^ T cells that were positive for LAG-3. (**B**,**C**) Cells (2 × 10^5^ per well) were cultured in vitro with GP33 peptide for 5 days, then analyzed by flow cytometry. (**B**) LAG-3 expression was determined by percentage of viable CD8^+^ T cells that were positive for LAG-3. (**C**) Proliferation of sub-populations of CD8^+^ T cells was determined by CFSE dilution. Proliferation score was calculated as in Ref. [31].

**Figure 6 cells-10-03445-f006:**
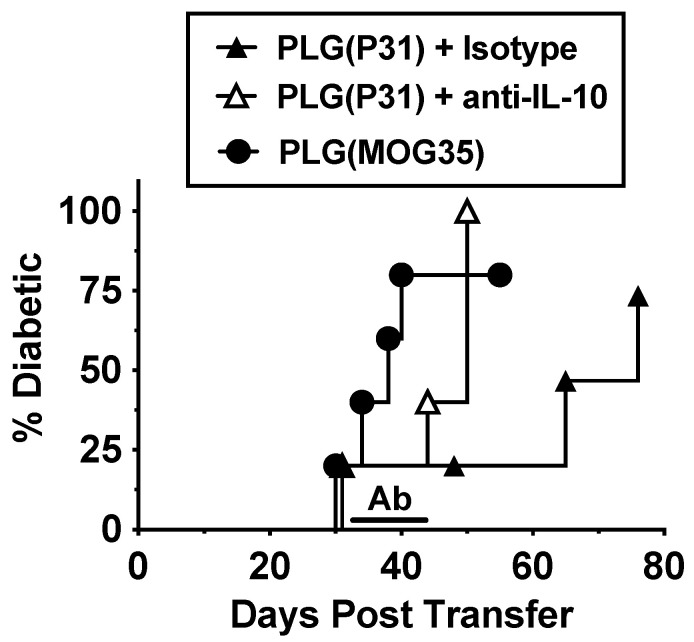
Tolerance in BDC-2.5 transfer model IL-10-dependent. NOD.*scid* mice were treated with either PLG(P31) or PLG(MOG35) within 24 h following adoptive transfer of 10 × 10^6^ P31-activated transgenic BDC-2.5 T cells and monitored for blood glucose levels. Mice were treated with isotype control or monoclonal anti-IL-10 (JES5-2A5) antibody from days 31 to 40 after transfer.

**Figure 7 cells-10-03445-f007:**
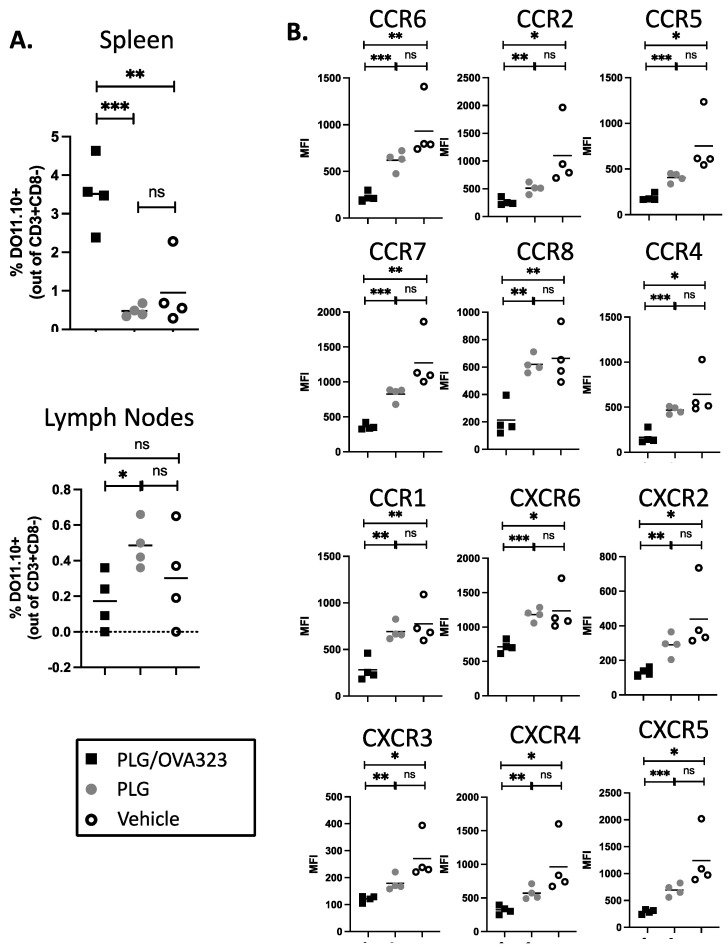
Antigen-loaded nanoparticles induce sequestration of transgenic CD4^+^ T cells in secondary lymphoid tissues and alter expression of chemokine receptors. Spleens were harvested from DO11.10 transgenic mice (*n* = 2–3) and CD4^+^ T cells were isolated by magnetic cell separation, then transferred to recipient BALB/C mice. Three days after cell transfer, mice were treated daily for 5 consecutive days with PLG/OVA323 or PLG. The numbers of DO11.10 T cells were determined in spleens or pooled inguinal, axillary, and brachial lymph nodes harvested 24 h after the final treatment. Statistical significance was determined by Student’s *t*-test. ns not significant, * *p* < 0.05, ** *p* < 0.01, *** *p* < 0.001. (**A**) Cell frequencies were determined by flow cytometry. (**B**) Levels of chemokine receptor expression on spleen-sequestered cells were determined by geometric mean fluorescence intensity.

**Figure 8 cells-10-03445-f008:**
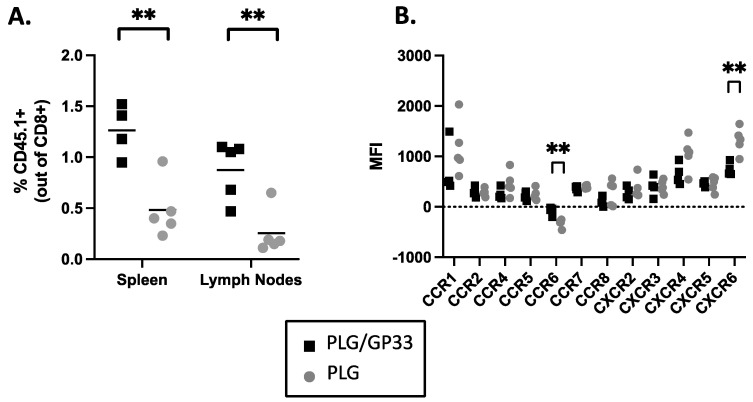
Antigen-loaded nanoparticles induce sequestration of transgenic CD8^+^ T cells in secondary lymphoid tissues and alter expression of chemokine receptors. Spleens were harvested from P14 transgenic mice (*n* = 2–3) and CD8^+^ T cells were isolated by magnetic cell separation, then transferred to recipient C57BL/6 mice. Three days after cell transfer, mice were treated daily for 5 consecutive days with PLG/GP33 or PLG. The numbers of CD45.1^+^ P14 T cells were determined in spleens or pooled inguinal, axillary, and brachial lymph nodes harvested 24 h after the final treatment. Statistical significance was determined by Student’s *t*-test. ** *p* < 0.01. (**A**) Cell frequencies were determined by flow cytometry. (**B**) Expression of chemokine receptors on spleen-sequestered cells was determined by geometric mean fluorescence intensity. Data are representative of three repeat experiments.

**Table 1 cells-10-03445-t001:** Properties of all nanoparticles used in this study.

Nanoparticle	Antigen	Transgenic Mouse Strain	Z-Average Diameter (nm) ^a^	Polydispersity Index ^a^	Zeta Potential (mV) ^a^	Antigen Load (μg/mg) ^b^
PLG	None	N/A	415.1 474.7 559.4	0.258 0.309 0.378	−93.6 +/− 11.7 −84.8 +/− 11.8 −59.7 +/− 7.83	N/A
PLG/GP33	LCMV GP 33–41	P14	496.4	0.276	−74.7 +/− 7.27	3.4
PLG/Ova323	Ovalbumin 323–339	DO11.10	522.5	0.376	−89.0 +/− 9.69	64.5
PLG(P31)	BDC-2.5 mimetope 1040-31	BDC-2.5	464.2	0.351	−84.1 +/− 10.1	2.1
PLG(MO35)	MOG 35-55	N/A	483.7	0.288	−84.7 +/− 14.3	5.3

Values are representative of typical batch characteristics. ^a^ Z-average diameter and zeta potential were determined by dynamic light scattering (+/− denotes standard deviation). ^b^ Antigen load was determined by CBQCA assay.

## Data Availability

The data presented in the current study are available upon request to the corresponding author.

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
