# Peer review of "Tolerance Induced by Antigen-Loaded PLG Nanoparticles Affects the Phenotype and Trafficking of Transgenic CD4+ and CD8+ T Cells"

_cells, 2021, doi:10.3390/cells10123445_

Round 1

Reviewer 1 Report

The research article entitled “Tolerance induced by antigen-loaded PLG nanoparticles affects the phenotypes and trafficking of transgenic CD4+ and CD8+ T cells” by Neef et al. to Cells. Research paper is fairly planned and experiments are rightly executed. The results are explained in a logical manner. I will recommend acceptance with minor revision. Please clarify the following concerns:

  1. Authors should provide the full name of PLD (i.e. poly(lactide-co-glycolide)) in its first appearance in the “Introduction” section.
  2. Introduction section lacks description of polymer based nanoparticles/nanocarriers in immunotherapy. Therefore, authors should edit the Introduction with some recent polymer-based nanoparticles in immune modulation related research/review papers. Some important refs:  Front. Bioeng. Biotechnol., 2020, 8, 1-20, J. Appl. Polym. Sci. 2020, 137, e49377, and Vaccines 2021, 9(8), 935.
  3. In section 2.5. “In vitro cell culture and secreted cytokine and proliferation”, please provide the name and details for the cell.
  4. Please provide the SEM/TEM image of PLG nanoparticles. The table 1 suggests the size distribution from 366 nm to 555 nm and antigen loading amount is 2,2 µg/mg to 64.5 µg/mg. Why is the loading capacity low for PLG (P31) and high for PLG/OVA323?
  5. How was the release profile of antigen from the nanoparticles?
  6. In Figure 1 C, there is no standard deviation (SD), why? How can you perform statistical analysis without triplicated data? If the data is triplicated (n=3) then there should be SD.

Author Response

We thank the reviewer for their comments and hope that their concerns are adequately addressed.

  1. Authors should provide the full name of PLD (i.e. poly(lactide-co-glycolide)) in its first appearance in the “Introduction” section.

We have rewritten the introduction and now the first mention of PLG includes the full name.

  1. Introduction section lacks description of polymer based nanoparticles/nanocarriers in immunotherapy. Therefore, authors should edit the Introduction with some recent polymer-based nanoparticles in immune modulation related research/review papers. Some important refs:  Front. Bioeng. Biotechnol., 2020, 8, 1-20, J. Appl. Polym. Sci. 2020, 137, e49377, and Vaccines 2021, 9(8), 935.

We have updated the introduction section to include references to other polymer-based nanoparticle papers, including a recently published review in which we summarized recent antigen-specific immunotherapies.

  1. In section 2.5. “In vitro cell culture and secreted cytokine and proliferation”, please provide the name and details for the cell.

In this section, we have now made it clear that the in vitro cell cultures are either whole spleens or whole lymph nodes.

  1. Please provide the SEM/TEM image of PLG nanoparticles. The table 1 suggests the size distribution from 366 nm to 555 nm and antigen loading amount is 2,2 µg/mg to 64.5 µg/mg. Why is the loading capacity low for PLG (P31) and high for PLG/OVA323?

We have added Figure 1, a pair of transmission electron micrographs that show the nanoparticles’ morphology. We still have no explanation for the extremely high antigen-load of PLG/Ova323 vs. the other nanoparticle types, but it is most likely not related to solubility as the same mass of peptide was fully dissolved in the coupling mixture. Hopefully, this will be answered with future studies.

  1. How was the release profile of antigen from the nanoparticles?

We did not perform experiments to address release profile of the antigen. We did, however, point to two previous studies, PMID: 27720992 and 28479234, that included data of these kinds.

  1. In Figure 1 C, there is no standard deviation (SD), why? How can you perform statistical analysis without triplicated data? If the data is triplicated (n=3) then there should be SD.

Statistical analysis was done using the three different age groups as replicates. We have updated the figure caption to specify this.

Reviewer 2 Report

The work of Tobias Neef et al. entitled “Tolerance Induced by Antigen-loaded PLG Nanoparticles Affects the Phenotype and Trafficking of Transgenic CD4+ and 3 CD8+ T Cells” presents a study of the effect of antigen-loaded nanoparticles on the phenotype and trafficking of transgenic CD4+ and CD8+ T cells for prevention and treatment of T1D. It is a well written manuscript and the topic is very pertinent. Although some interesting data have been presented, several issues need to be addressed and added before this manuscript could be considered for publication. I would recommend a resubmission with minor revision based on the following general comments:

  1. It is missed information about the reactants and exactly methodology used for the synthesis of each PLG nanoparticles. Please could you add more information about single and double emulsion processes.
  2. Regarding the conjugation between the PLG nanoparticle and the peptides, have you checked by any technique like NNM or IR-ATR the amide formation to confirm the conjugation?
  3. Which is the molar ratio between the PLG and peptide-coupled-PLG?
  4. Are the authors checked the morphology of peptide@PLG nanoparticles by microscopy?
  5. It has been shown that the tolerogenic efficiency of antigen loaded PLG nanoparticles can be highly variable based on batch. Have you tested batch variability in your experiments? This is very important point for using nanoparticles as a reliable therapeutic tolerogenic agent.
  6. Table 1. Please add standard deviation to all data. Size mean value should be accompanied by the Polydispersity Index (PdI), which is a measure of the heterogeneity of a sample based on size.
  7. The major drawbacks in the development of peptide-PLG based nanoparticles are the high initial burst, incomplete release and instability of the encapsulated peptides. Initial burst release means the rapid release of a large amount of encapsulated peptide. Have you checked the stability of your PLG particles in vitro upon time?
  8. Line 211-212. The observed variations in terms of size and zeta for the five prototypes is mainly due to the different peptide-molecules being trapped, which modify nanoparticle size and surface charge, more than a batch-to-batch variation.
  9. The antigen loading for PLG/GP33, PLG/P31 and PLG/MOG35 is very low in comparison to PLG/OVA323. Please do you have any explanation for this fact. Can it be related to peptide solubility in PBS?
  10. Line 100. Revise spelling: “with the with their”.

Author Response

We thank the reviewer for their comments and hope that their concerns are adequately addressed.

  1. It is missed information about the reactants and exactly methodology used for the synthesis of each PLG nanoparticles. Please could you add more information about single and double emulsion processes.

We have not added much detail regarding the synthesis of single and double emulsion nanoparticles because these can be found in older publications. The revised version makes it clearer that the same procedure was used

  1. Regarding the conjugation between the PLG nanoparticle and the peptides, have you checked by any technique like NNM or IR-ATR the amide formation to confirm the conjugation?

We did not perform experiments to address burst release or confirm peptide to polymer conjugation. We did, however, point to two previous studies, PMID: 27720992 and 28479234 that included data of these kinds.

  1. Which is the molar ratio between the PLG and peptide-coupled-PLG?

We have added details regarding the ratio between peptide-coupled and uncoupled PLG in the Methods section, although it was only possible to indicate amounts in terms of milligrams instead of moles.

  1. Are the authors checked the morphology of peptide@PLG nanoparticles by microscopy?

We have added Figure 1, a pair of transmission electron micrographs that show the nanoparticles’ morphology.

  1. It has been shown that the tolerogenic efficiency of antigen loaded PLG nanoparticles can be highly variable based on batch. Have you tested batch variability in your experiments? This is very important point for using nanoparticles as a reliable therapeutic tolerogenic agent.

We have pointed out that batch-to-batch variation is accounted for by determining antigen load of each batch via CBQCA assay and using only those that contain enough antigen.

  1. Table 1. Please add standard deviation to all data. Size mean value should be accompanied by the Polydispersity Index (PdI), which is a measure of the heterogeneity of a sample based on size.

We have updated Table 1 to include polydispersity indexes and standard deviations. We also included properties for three different batches of PLG particles. We expect this will satisfy questions about batch-to-batch variability and make clear that all nanoparticle types fall within a typical range of diameters and zeta potentials, regardless of the identity of the antigen that was loaded.

  1. The major drawbacks in the development of peptide-PLG based nanoparticles are the high initial burst, incomplete release and instability of the encapsulated peptides. Initial burst release means the rapid release of a large amount of encapsulated peptide. Have you checked the stability of your PLG particles in vitro upon time?

We did not perform experiments to address burst release or confirm peptide to polymer conjugation. We did, however, point to two previous studies, PMID: 27720992 and 28479234, that included data of these kinds.

  1. Line 211-212. The observed variations in terms of size and zeta for the five prototypes is mainly due to the different peptide-molecules being trapped, which modify nanoparticle size and surface charge, more than a batch-to-batch variation.

We have updated Table 1 to include polydispersity indexes and standard deviations. We also included properties for three different batches of PLG particles. We expect this will satisfy questions about batch-to-batch variability and make clear that all nanoparticle types fall within a typical range of diameters and zeta potentials, regardless of the identity of the antigen that was loaded.

  1. The antigen loading for PLG/GP33, PLG/P31 and PLG/MOG35 is very low in comparison to PLG/OVA323. Please do you have any explanation for this fact. Can it be related to peptide solubility in PBS?

We still have no explanation for the extremely high antigen-load of PLG/Ova323 vs. the other nanoparticle types, but it is most likely not related to solubility as the same mass of peptide was fully dissolved in the coupling mixture. Hopefully, this will be answered with future studies.

  1. Line 100. Revise spelling: “with the with their”.

We thank the reviewer for pointing this out and have corrected the error.